# Antarctic Soil and Viable Microbiota After Long-Term Storage at Constant −20 °C

**DOI:** 10.3390/biology14030222

**Published:** 2025-02-20

**Authors:** Cristian-Emilian Pop, Sergiu Fendrihan, Nicolai Crăciun, Garbis Vasilighean, Daniela Ecaterina Chifor, Florica Topârceanu, Andreea Florea, Dan Florin Mihăilescu, Maria Mernea

**Affiliations:** 1Non-Governmental Research Organization Biologic, 14 Schitului Str., 032044 Bucharest, Romaniavasilighean@gmail.com (G.V.); chifor.daniela-ecaterina@bio.unibuc.ro (D.E.C.); andreeaflorea708@gmail.com (A.F.); 2Department of Anatomy, Animal Physiology and Biophysics, Faculty of Biology, University of Bucharest, 91–95 Splaiul Independenței Str., 050095 Bucharest, Romania; dan.mihailescu@bio.unibuc.ro (D.F.M.); maria.mernea@bio.unibuc.ro (M.M.); 3National Commission for Antarctic Research of the Romanian Academy, 125 Calea Victoriei, 010071 Bucharest, Romania; antarctic.tf2021@gmail.com; 4Zoology Section, Department of Biochemistry and Molecular Biology, Faculty of Biology, University of Bucharest, 91–95 Splaiul Independenței Str., 050095 Bucharest, Romania; nicolai.craciun@bio.unibuc.ro; 5Stefan S. Nicolau Institute of Virology, 285 Mihai Bravu Ave., 030304 Bucharest, Romania

**Keywords:** east continental Antarctica, Antarctic soil, long-term storage, Antarctic microbiota, cold resilience

## Abstract

Soil samples taken from three locations in east continental Antarctica were airtight sealed and stored at −20 °C for over a decade before being analyzed. Interestingly, some microbial strains survived the long-term storage while isolated from the habitat. Soil organic and elemental contents were also analyzed and the results varied significantly, although Fe was the predominant element in all samples. Apostrophe Island presented particularly higher organic matter content, and higher levels of Co, Cr, Ni, Sr, Cu and P, as well as As, which was detected in significant quantities. The arsenic soil samples presented viable *Pseudomonas arsenicoxydans*, and other preserved and recovered strains included *Pseudomonas graminis*, *Sporosarcina aquimarina*, and *Geomyces pannorum*, extremophiles which we can now consider beyond any reasonable doubt to be facultative psychrophiles which can form dormant states.

## 1. Introduction

A finite limit on which life can remain active is bordered by cold temperatures. To date, the coldest reported temperatures for microbial metabolism and growth are −12 °C for the sea ice bacterium *Psychromonas ingrahamii*, and −33 °C for the Antarctic glacial isolates *Paenisporosarcina* sp. B5 and *Chryseobacterium* sp. V3519-10 [1,2]. There is relatively little known about the microbial life in Antarctic permafrost compared to that in Arctic and Alpine permafrost [3], yet it is generally accepted that biological activity at such freezing temperatures is conditioned by the stress of high solute concentrations, and a decrease in molecular motion and energetics. Many places on Earth, and potential environments for life elsewhere in the solar system, have subfreezing temperatures; therefore, it is of considerable interest to know which microorganisms are the most versatile and best-adapted psychrophiles [1]. Therefore, the microorganisms residing in Earth’s cryoenvironments are ideal subjects for investigating the limits of life at low temperatures [4,5]. These environments are habitats for diverse microbial communities, some extremophiles and other facultative psychrophiles, with studies demonstrating their metabolic activity below 0 °C in pure cultures [6,7], as well as in soil microcosms [8,9].

Permafrost experiences constant subzero temperatures (typically between −10 and −20 °C), although due to global warming, temperatures up to 0 °C have also started to be recorded, with an annual increase predicted [10,11]. Interestingly, only about 0.1% of Antarctica is ice-free soils [12]; these are under very harsh environmental conditions, yet they contain bacterial and archaeal phyla. Some parts of the Antarctic are dominated by eukaryotes, but areas containing cold deserts without snow contain microbiota dominated by bacteria, as investigated in the frame of Antarctic Conservation of Biogeographic Regions [13], which discovered that there is an overlap between the community structure of Antarctic maritime soils and continental soil communities. Furthermore, it was suggested that bacterial communities might be impacted by regional climatic and other environmental changes.

Other authors [14] showed that the microbial communities have a certain heterogeneity despite the limiting environmental factors, acting with better survival strategies. In the context of climate change, changes in the composition of microbial communities take place in relation to higher-rank taxa and the abundance of fungi and bacteria [15], with the shifts toward generalist bacterial communities following warming weakening the linkage between the bacterial taxonomic and functional richness.

As Antarctica is less influenced by human impact due to being hardly accessible, there are probably still many undiscovered organisms. In recent years, amplicon-based next-generation sequencing (NGS) studies have allowed fast and thorough examinations and identification of unknown species [16]. Even so, many groups of bacteria in the Antarctic depend on factors such as temperature variations, pH, and even the existence of bird colonies and vegetation; the moisture content is also a limiting factor for some communities of microorganisms [17].

At the same time, although considered the least human-impacted area, the microbial communities from Antarctica present genes related to resistance to antibiotics, metals, and biocides [18]. Such representative strains are *Polaromonas*, *Streptomyces*, *Bulkhoderia,* and *Pseudomonas*, probably consisting of a natural-origin resistome, which could transfer to pathogenic bacteria. Of particular interest and concern is how and when the rise in temperature will change the soil microbiome, along with the cascade of events that this can trigger.

In this work, we aimed to evaluate Antarctic soils from a chemical as well as microbiological point of view after almost 13 years of separation from the environment, using samples stored at a constant −20 °C. Furthermore, we also addressed the soil particularities of non-organic content, parameters which should have remained the same from the time of collection.

## 2. Materials and Methods

### 2.1. Sampling Locations, Storage, and Handling

Soil samples were collected during December 2010–January 2011 in the East Antarctic coastal region (Table 1) and sealed in aseptic bags. The samples were stored undisturbed in airtight plastic containers at −20 °C until their opening in June 2024, as they were made available long after the passing of the researcher in charge, to whom this work is dedicated.

The soil samples were processed in sterile conditions; sealed sample bags were sterilized with anhydrous ethanol on their exterior, defrosted, opened, and aliquoted in a class 2 microbiological safety cabinet. Element analysis was performed via inductively coupled plasma atomic emission spectroscopy (ICP-AES) and further soil analysis was performed with Fourier-transform infrared spectroscopy coupled with attenuated total reflection (FTIR-ATR); the remaining aliquots were used for microbial metabolic pattern and species identification.

### 2.2. ICP-OES Soil Analysis

Determination of 21 elements was performed via ICP-AES (Agilent 5100 and 5900 ICP-OES, Santa Clara, CA, USA), and the concentrations of the compounds were determined using stoichiometric calculations. Samples were homogenized by crushing, grinding, and pulverization, before being dried and mineralized in aqua regia. Gravimetric determination of dry matter and determination of moisture were performed (CSN ISO 11465, CSN EN 12880, CSN EN 14346:2007, CSN 46 5735) and for each sample, 1 g was dried and then mixed with 10 mL of aqua regia.

After 16 h, the mixture was heated in graphite hot blocks for 2 h at 130 °C; the solution was then filled up to a volume of 50 mL and filtered through a filter of 0.45 μm porosity before analysis by the ICP technique (US EPA Method 200.7, CSN EN ISO 11885, US EPA Method 6010, SM 3120). The analysis was performed by an ISO/IEC 17025 accredited external service provider (Measurlabs, Helsinki, Finland); the raw data report of the analysis is presented in Appendix A.

### 2.3. FTIR-ATR Soil Evaluation

Ten soil sample aliquots were dried prior to FTIR spectroscopy measurements. A Bruker Tensor 27 spectrometer (Bruker Optik GmbH, Ettlingen, Germany) equipped with an attenuated total reflection (ATR) module was used to collect the spectra (4000–400 cm^−1^ range, 2 cm^−1^ resolution). The peaks characteristic to each spectrum were determined using OPUS software (version 7.2, Bruker Optik GmbH, Ettlingen, Germany). Peak assignment was performed based on the literature.

Spectra were compared using calculated Euclidean distances (EDs) as a metric [19]. This metric counts the differences between two spectra, being given by the formula$$d\left(S_{i},S_{j}\right)=\sqrt{\sum_{k=1}^{N}{\left(S_{i,k}-S_{j,k}\right)}^{2}}$$
where $S_{i,k}$ and $S_{j,k}$ are the absorbance values at wavenumber index *k* for samples *i* and *j*, while *N* is the total number of wavenumbers in the dataset [19,20,21]. As previously stressed by other authors [19], the method is sensitive to differences in offset and sloping of the baseline, as well to differences in absolute values of the absorbance. Thus, prior to the analysis, we corrected the offset and aligned the baseline of the spectra across all samples. Some differences in the absolute absorption of samples still remain, but these are negligible in comparison to the variations in the overall shape of the spectra.

The calculation of pairwise EDs of all spectra resulted in an ED matrix that was further analyzed by hierarchical agglomerative clustering (HAC) [20,22] in order to identify similar spectra. HAC is a “bottom up” approach to building a cluster tree that starts with each spectrum as a single cluster and successively merges pairs of clusters based on the distance metric until a single cluster remains [20,22]. Visualization of the cluster tree is performed by a dendrogram in which each leaf represents a sample, the nodes represent obtained clusters, and the horizontal axis indicates the ED [20].

### 2.4. Microbial Metabolic Pattern

For each sample, 1 g was mixed in sterile conditions with 14 mL of liquid Amies media (Rmbio, Missoula, MT, USA), shaken at 200 rpm for 2 h at room temperature, left to settle for 10 min, and then the supernatant was inoculated on Biolog Ecoplates (Harvard, Cambridge, MA, USA), which are considered to be one of the most suitable methods for evaluating live bacterial community compositions [23,24], as we aimed to investigate viable cells only. The Biolog Ecoplates are 96-well plates that hold 31 carbon assays in triplicate, which are split into 6 groups: polymers, carbohydrates, carboxylic acids, amino acids, amines, and phenolic compounds. After inoculations, the plates were sealed with tape, incubated at 15 °C, and read for absorbance every 24 h at 590 nm (FlexStation Multimode Plate Reader, Molecular Devices, San Jose, CA, USA) for 96 h, according to the manufacturer. After 96 h, some wells developed completely (full substrate usage), while others presented no growth whatsoever.

### 2.5. Primary and Final Taxonomic Identification

The content of each Biolog Ecoplate well that developed microbial growth, observed via absorbance at 590 nm, was used to inoculate nutrient agar plates (5 g/L peptone, 5 g/L NaCl, 2 g/L yeast extract, 1 g/L meat extract), and for fungal isolation, Sabouraud with chloramphenicol agar plates (10 g/L meat and casein peptic extract, 40 g/L dextrose monohydrate, 0.05 g/L chloramphenicol) was used; all culture media were obtained from MLT, Romania. The agar plates were incubated for 96 h at 15 °C, and the resulting colonies were isolated on fresh plates, followed by a primary identification on chromatic detection agar. Pure cultures no older than 96 h were harvested and suspended in 300 μL nuclease-free water (Thermofisher, Waltham, MA, USA); in the case of fungal specimens, a sterile micropestle was used in grinding to assist in evenly suspending the organism in the water. For each suspension obtained, 900 μL of anhydrous ethanol (Sigma-Aldrich, St. Louis, MO, USA) was added and the samples were subjected to BacSeq^®^ and FunITS^®^ 16S sequencing, performed by a cGMP-compliant, ISO 17025-accredited external service provider (Accugenix Charles River, Kaarst, Germany); the consensus sequences are presented in Appendix A.

The BacSeq^®^ analysis represents the DNA sequencing of the 16S ribosomal RNA gene in bacteria and the FunITS^®^ analysis is the comparative sequencing of the ribosomal DNA ITS2 region in fungi. The workflow of selected analyses involves several steps: (i) Genomic DNA extraction from samples. (ii) Amplification of target sequences using polymerase chain reaction (PCR) and the relevant primers which are the first 500 base pairs of the 16S rRNA gene for bacteria (527R reverse and 005F forward primers) and the 350 bp ITS2 region for fungi (ITS3 forward and ITS4 reverse/ITS86F forward and ITS4 reverse primers). The PCR product was purified using a magnetic bead solution to remove any excess primers and unincorporated deoxynucleotides. (iii) Cycle sequencing of the PCR product using dye terminators. Fluorescently labeled dideoxynucleotides are incorporated into DNA fragments up to 500 bases. These are purified and analyzed using an automated fluorescent sequencer. DNA fragments migrate through a capillary, where a laser excites their fluorescent labels. A CCD camera captures the signals, generating an electropherogram that represents the nucleotide sequence. (iv) DNA sequence filtering based on quality. (v) Strain identification based on screening against the validated Accugenix^®^ library and phylogeny assessment.

### 2.6. Statistical Analysis

The correlations between the concentrations of elements detected in the soil were calculated, and plotting of resulting data was performed with the Advanced Data Analysis tool of ChatGPT [25,26]. The elements with concentrations below the detection limit (Ag, Hg, Mo, Sb, Sn, and Tl) were excluded from the analysis. The obtained correlations were analyzed by the authors of this paper.

## 3. Results

### 3.1. Soil Physico-Chemical Characterization

The FTIR spectra that were measured on the ten samples were compared and clustered based on Euclidean distance values calculated between pairs of spectra. The plot in Figure 1a shows the resulting two clusters, one comprising the spectra of samples 1 to 7 and one comprising the spectra of samples 8 to 10. In the case of the first cluster, we identified two subclusters, samples 1–4 and samples 5–7, that show a small distance. The two subclusters were analyzed separately.

The spectra of samples 1 to 4 are presented in Figure 1b; these spectra have a similar shape and corresponding absorption peaks. The assignment of peaks was performed based on the work of Volkov et al. [27], who used FTIR to characterize silicate soils. Several peaks can be attributed to the presence of SiO_2_: 430 cm^−1^ (Si-O deformation), 463 cm^−1^ (O-Si-O bend), 521 cm^−1^ (silicate O-Si-O bend), 644 cm^−1^ (Si-O, sulfate, bentonite), 694 cm^−1^ (Si-O-Si bend in SiO_2_), 777 cm^−1^ (Si-O-Si), 797 cm^−1^ (lattice symmetrical Si-O-Si stretch specific to SiO_2_), 1001 cm^−1^ (Si-O lattice stretch specific to SiO_2_), and 1080 cm^−1^ (O-Si-O lattice stretch). Some peaks are evidence of the presence of organic matter, like the peak at 1080 cm^−1^ occurring in the spectra of samples 1 and 2, which can be assigned to C-N vibration. Two other weak bands centered around 1649 cm^−1^ and 1543 cm^−1^ represent the amide I and amide II bands specific to proteins [28].

The amide bands specific to proteins are missing from the spectra of samples 5–7 (Figure 1c). In their case, we notice a strong absorption band centered at ~997 cm^−1^, which is the Si-O-Si band [27,29]. Some smaller absorption peaks between 800 and 400 cm^−1^ can also be assigned to Si-O-Si symmetrical and asymmetrical bending specific to quartz [30]. A broad absorption band found between 3500 and 3000 cm^−1^ can be assigned to O-H stretching vibrations from water molecules coordinated by the silica in the sample and to silanol groups (Si-OH) found at the surface of silica particles in sand [29]. Water molecules coordinated at the surface of silica particles give an additional absorption band centered at ~1650 cm^−1^ [29].

The spectra in the last cluster, samples 8–10 (Figure 1d), present strong amide bands, centered at ~1634 cm^−1^ and ~1547 cm^−1^. Other peaks show the presence of organic material, like those around 1379 cm^−1^, 1418 cm^−1^, 1452 cm^−1^ (CH_2_, CH_3_ group deformation), 2853 cm^−1^, and 2922 cm^−1^ (C-H stretch of CH_2_ group) [31]. The peaks specific to silica are also present at around 417 cm^−1^ (Si-O deformation), 519 cm^−1^ (silicate O-Si-O bend), 534 cm^−1^ (α quartz), and 1030 cm^−1^ (Si-O stretching) [27]. The water coordinated by the silica particles contributes to the spectrum with the strong absorption band between 3700 and 3000 cm^−1^ and the band overlapping the amide I band at ~1634 cm^−1^ [29].

The analysis above shows that the soil in all samples comprises silica and more or less organic matter. The clustering of spectra showed the presence of two groups, one with little (subgroup of samples 1–4, Camp Faraglione) or undetectable (spectra of samples 5–7, Edmonson Point) organic matter and one with more organic matter (spectra of samples 8–10, Apostrophe Island). Thus, the three subclusters coincide with the three locations from where the samples were collected, supporting a difference between the soil content of these locations. The difference will be further described based on the analysis of the metals present in the soil.

ICP-OES element analysis revealed that the most prominent element in all samples was Fe, and samples with high organic matter content (samples 8–10, Apostrophe Island) presented higher contents of Co, Cr, Ni, Sr, and especially Cu and P; interestingly, As was also detected in significant quantities (Table 2).

### 3.2. Soil Physico-Chemical Correlations

The correlations between the concentrations of elements were investigated depending on location. The plots in Figure 2 show that the correlations between element concentrations present significant variations in the tree locations.

In the first location (Camp Faraglione, coordinates 74°42′57″ S–164°06′41″ E), we observed mostly strong positive correlations between the concentrations of elements, especially for As, Co, Cr, Cu, Fe, Li, Mn, P, Pb, Sr, V, and Zn. The concentration of Be presents fewer strong positive correlations with Cr, Pb, and Sr. There is only a strong negative correlation between Ba and Be.

In the second location (Edmonson Point, coordinates 74°18′47.1″ S–165°04′10.2″ E), the number of strong positive correlations is lower, involving elements like Ba or Zn. There are perfect positive correlations between the concentrations of Be with Li or Fe, Co with Cu, Li with Fe, P with Sr or V, and Sr with V. In contrast to the first location, in the second one, we observed strong negative correlations between the concentration of As and the concentrations of all the other elements except for Cd, Pb, Sr, and Zn. The concentration of Pb is strongly negatively correlated with the concentrations of Co, Cu, P, Sr, and V.

In the third location (Apostrophe Island, coordinates 73°31.134′ S–167°25.984′ E), we obtained a different pattern of correlations. There are some strong positive correlations, from which we mention the perfect positive correlations between the concentrations of As with P, Sr, and Zn; Ba with Fe; Cd with Zn; Cr with Fe; Li with V; and P with Sr and Zn. The number of strong negative correlations is even larger, involving most elements. The perfect negative correlations are seen between the concentrations of As with Ba and Fe; Ba with P, Sr, and Zn; Cr with P and Sr; and Fe with P, Sr, and Zn.

These results show that there are clear differences between the concentrations of elements in the three locations. For instance, the concentration of As presents strong positive correlations with the concentrations of other elements in location 1, strong negative correlations with the concentrations of other elements in location 2, and strong positive and negative correlations with the concentrations of other elements in location 3. In the case of Ba, there was only a strong negative correlation between its concentration and that of Be in location 1, more strong positive correlations with the concentrations of other elements in location 2, and both strong positive and negative correlations in location 3. The concentration of Pb presents only strong positive correlations in location 1, only strong negative correlations in location 2, and no strong correlations in location 3. The situation is similar for other elements; therefore, we tested the consistency of these correlations in all three locations by calculating the correlations over all the data regardless of location.

The correlations between the element concentrations detected in all the samples are shown in Figure 3. The plot shows that there are strong positive correlations (>0.7) between the concentration of As with Be, Co, Cu, P, and Sr; Ba with Li, Pb, and V; Be with Ce, Cu, P, and Sr; Co with Cu; P with Sr; Cu with P and Sr; Fe with Li and Pb; Li with Pb and V; P with Sr; and Pb with V. A strong negative correlation (r < −0.7) is seen between the concentrations of Cr and Mn.

### 3.3. Soil Microbiota

The average metabolic response analysis of Biolog Ecoplates, which is calculated as the average of the mean difference between the optical densities of the carbon source-containing wells and the control wells [32,33], revealed that the most notable substrate utilization occurred for locations 2, 6, and 9. Phenolic compound (2-Hydroxy Benzoic Acid, 4-Hydroxy Benzoic Acid) utilization had the best metabolic response, compared to polymers (Tween 40, Tween 80, alpha-Cyclodextrin, Glycogen), which were the least preferred carbon sources (Figure 4).

We further used all Biolog Ecoplates wells that developed microbial growth to recover microbiota. All well contents were transferred onto nutrient media plates, resulting in colonies which were isolated on fresh media plates and then identified via 16S rRNA gene analysis using the AccuGENX-ID Database (consensus sequences are presented in Appendix A). The percentage differences in the sequence alignment between the unknown species and each individual library entry, as well as the neighbor-joining trees, are presented in Appendix A. The sample identified as *Pseudomonas graminis* has a 0.0% difference in the sequence alignment compared to the library entry; *Geomyces pannorum* has a 0.30% difference, *Pseudomonas arsenicoxydans* 0.10%, and Sporosarcina aquimarina a 1.59% difference.

In terms of colony morphology and species distribution, sample 1 developed only circular and slightly convex yellow colonies, up to 5 mm in diameter with a regular smooth edge, identified as *Pseudomonas graminis*, and sample 2 developed both *P. graminis* and the white filamentous fungi *Geomyces pannorum*. In sample 4, only *G. pannorum* was present. Sample 6 contained circular colonies that were either small or punctiform, of cream color, identified as *Pseudomonas arsenicoxydans*, which was also present in sample 8, along with *P. graminis*, and the light-orange-colored mucoid colonies of *Sporosarcina aquimarina.* Sample 9 presented only *P. arsenicoxydans,* and no colonies were observed for samples 3, 5, 7, and 10.

## 4. Discussion

The physico-chemical analysis of soil samples from the three distinct locations—Camp Faraglione, Edmonson Point, and Apostrophe Island—revealed significant differences in their silica and organic matter content, as well as variations in metal concentrations and their inter-element correlations.

FTIR spectroscopy identified silica as a common component across all samples, with varying levels of organic matter. Soils from Camp Faraglione (samples 1–4) showed low organic content, while Edmonson Point soils (samples 5–7) revealed minimal organic signatures but prominent Si-O bands. In contrast, Apostrophe Island soils (samples 8–10) were rich in organic markers, including amide bands indicative of proteins and CH group vibrations, alongside silica-associated peaks; these contents seem to have no relation with the microbial abundance.

The clustering of FTIR spectra reflected these compositional differences, corresponding to the geographic origins of the samples. Further analysis of metal concentrations highlighted that samples with higher organic matter content (e.g., Apostrophe Island) also had elevated levels of Co, Cu, P, and As, suggesting that organic content influences metal retention.

Correlation analysis uncovered unique elemental interrelationships at each site. Camp Faraglione demonstrated mostly positive correlations, particularly among As, Co, Cr, and Cu. Edmonson Point showed a mix of positive and negative correlations, with a notable negative relationship between As and most other elements. Apostrophe Island presented a complex pattern of strong positive and negative correlations, emphasizing the influence of its higher organic and metal content.

Within the soil samples, viable microbiota was investigated and detected; the ability of the identified strains to survive after almost 13 years of frozen storage at −20 °C, without cryoprotectants, highlights their resilience and possible potential for dormancy. The findings align with previous studies suggesting that bacterial and fungal communities in permafrost and cold soils can persist for extended periods in a metabolically inactive or dormant state [34].

There are few reports of bacterial survival after long-term storage at −20 °C in raw samples, without glycerol or other protective/cryogenic agents. In a 44-month-long study on the effects of frozen storage (at −20 °C) on soil culturable microorganism viability, it was observed that microbial populations significantly decreased after 8–12 months of frozen storage, but the changes did not, however, significantly affect the total culturable microorganism count or differences in microbial biodiversity between soils. Furthermore, it was recommended that frozen storage for up to 8 months may be allowed for analyses of culturable microbial biodiversity, and some longer times may be reasonably acceptable [35]. However, the viability of the sporulating bacillus and fungus populations significantly decreased after 8–12 months of frozen storage. In our study, among the four microbial strains found viable, only one was fungi, *G. pannorum;* thus, this strain stands out as particularly resilient to long-term frozen storage.

In a study on microbial metabolic diversity in east continental Antarctica, the researchers mentioned the fact that the utilization of Tween 40, Tween 80, D-galacturonic acid, L-asparagine, D-mannitol, and L-arginine contributed to the diversity in the microbial metabolic profile at different sites [36]. It is essential to mention that Antarctic taxonomic richness is not yet fully understood; ref. [37] investigated 62 samples of Antarctic soils from all categories from Northern Victoria Land, and found species of *Bacillus*, *Brevibacillus*, *Paenibacillus,* and others, as well as a strain of *Bacillus thuriginesis* even in non-visited and uncontaminated places, suggesting that the organisms were living in these soils rather than being contaminants; in their study, a representative strain from Apostrophe Island was identified as *B. thuriginesis* serovar pirenaica (H57), the same as one isolated from a dust sample collected from a corn silo in Elorz, Spain [38,39].

As previously mentioned, *G. pannorum* was the only fungal strain observed in our study; the species is known as keratinolythic and had been previously isolated from Antarctic samples [40]. Its outstanding resilience has been reported including in regard to its adaptation to cryopegs (permafrost overcooled water brines), in which *G. pannorum* thrives significantly better than other fungi [41]. Another interesting strain detected was *P. graminis*, which was first described in 1999 after being identified in the phyllosphere of grasses, not able to reduce nitrate to nitrite, yet able to utilize a wide range of compounds individually as a sole carbon source [42]. To the best of our knowledge, although the genus is common in Antarctica [43,44], this is the first report of the species being found in Antarctic soil.

The arsenite-oxidizing bacterial strain *Pseudomonas arsenicoxydans* was first isolated from the Atacama Desert, from an arsenic-polluted site where arsenic content varied (≈1100 µg L^−1^ for water, and ≈550 µg L^−1^ in sediments) [45]. The species was experimentally evaluated to be able to withstand 5 mM of arsenic (374 mg/L), and fully oxidize the arsenite present in the medium after 48 h incubation with lactate as a carbon source [45]. Our highest arsenic content was detected in sample 9 from Apostrophe Island (5.27 mg/kg), and only *P. arsenicoxydans* was detected in the sample. Although many *Pseudomonas* spp. have been reported to grow in arsenic-rich environments [46,47], sample 9 with the highest arsenic content contained only *P. arsenicoxydans* and no *P. graminis*. Still, *P. graminis* was present in sample 8 (a lesser high-content arsenic soil), only along with *P. arsenicoxydans.* This observation strengthens the presence of *P. arsenicoxydans* as an arsenite-oxidizing strain; however, we could not find any reports of *P. graminis* related to growth in arsenic media [48]. It is important to mention that due to the imperfect sequence alignment compared to the library entry (0.10% difference), it is possible that a *biovar.* or highly similar strain of *P. arsenicoxydans* with the same arsenic metabolic competence was detected in this study. In a soil metal and microbiota study, Lin et al. [49] stated that few connections were observed between microbial communities and arsenic; however, in our case, there is an evident correlation between the high arsenic content and *P. arsenicoxydans.* The study also reported that bioavailable Fe and Mn were recognized as the major driving forces shaping the taxonomic structure of microbial communities, yet this referred strictly to the bioavailable fractions.

Of particular interest is *S. aquimarina*, a facultative anaerobe halophile isolated from seawater in Korea in 2001 [50]. Unlike *Sporosarcina antarctica*, a related species previously identified in Antarctic soils collected off King George Island, West Antarctica [51], *S. aquimarina* has also not been reported in Antarctica until now, to the best of our knowledge. According to the neighbor-joining tree (Appendix A), the detected *S. aquimarina* is closely related to *S. koreensis*, another soil-isolated microorganism [52]. An interesting study [53] reported that at cold temperatures, supplementation with leucine positively influenced the growth rates of *S. koreensis* but had no effect on *S. aquimarina*, and that these bacteria modulate their fatty acid compositions in response to the growth environment.

The strains of microorganisms identified in this work seem to be facultative psychrophilic rather than real psychotrophs, and they are oligotrophic, adapted to both mild and harsh environmental conditions. Their resistance to −20 °C for so many years can be appealing for future environmental studies in extreme conditions.

Antarctica’s average annual temperature according to data provided by the Australian Antarctic Program [54] ranges from −10 °C to −60 °C on the coast; however, the temperature can exceed +10 °C in summer and fall to below −40 °C in winter. On the elevated inland, the temperature can fall below −80 °C in winter and −30 °C in summer.

Such environmental extremes can induce subtle yet critical metabolic traits which are present in the native microbiota and might not be found anywhere else, aspects which require further detailed studies.

## 5. Conclusions

The present study brings new insights regarding not only the physico-chemical traits of Antarctic soils of Camp Faraglione, Edmonson Point, and the least-known location Apostrophe Island, but also provides valuable data on some of the native microbiota of these locations. With the limitation that only aerobic and facultative anaerobic species were investigated, the identified bacteria *Pseudomonas graminis*, *Pseudomonas arsenicoxydans,* and *Sporosarcina aquimarina*, as well as the fungus *Geomyces pannorum*, have particular viability and resistance to long-term cold storage after isolation from their environment, which represents another glimpse into the microbial diversity and resilience in the harsh Antarctic environment.

## Figures and Tables

**Figure 1 biology-14-00222-f001:**
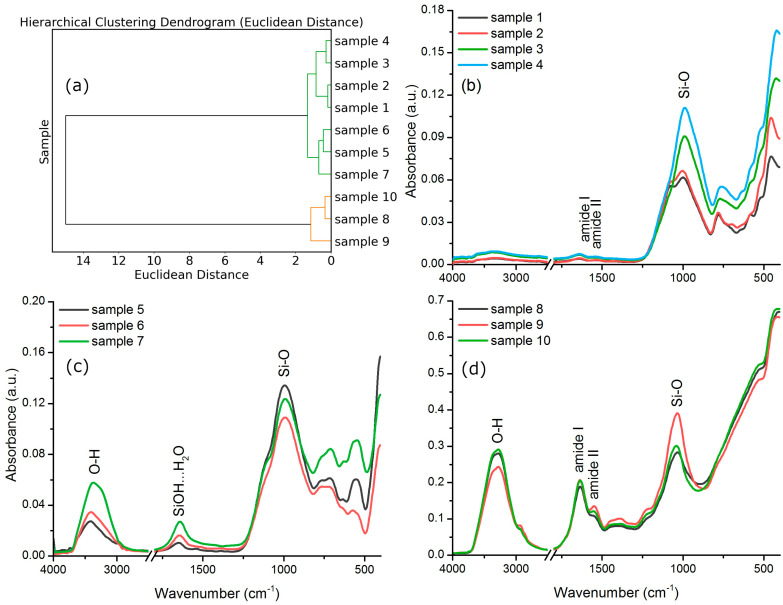
(**a**) Hierarchical clustering dendrogram of spectra calculated based on pairwise Euclidean distances. (**b**) ATR-FTIR spectra of samples 1–4. (**c**) ATR-FTIR spectra of samples 5–7. (**d**) ATR-FTIR spectra of samples 8–10.

**Figure 2 biology-14-00222-f002:**
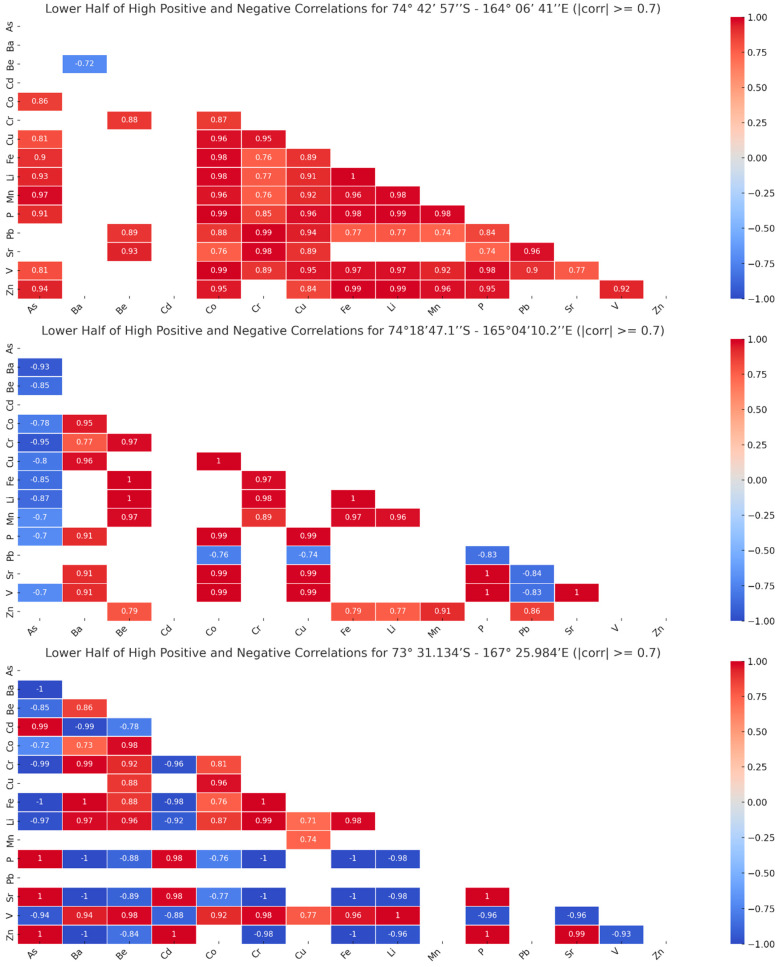
The lower half of high positive and negative correlations (the module of the correlation value is larger than or equal to 0.7) between the concentrations of elements in each station. The high positive correlations are represented in shades of red, while the high negative correlations are represented in shades of blue. Locations are identified based on their coordinates: Camp Faraglione—top plot; Edmonson Point—middle plot; Apostrophe Island—bottom plot.

**Figure 3 biology-14-00222-f003:**
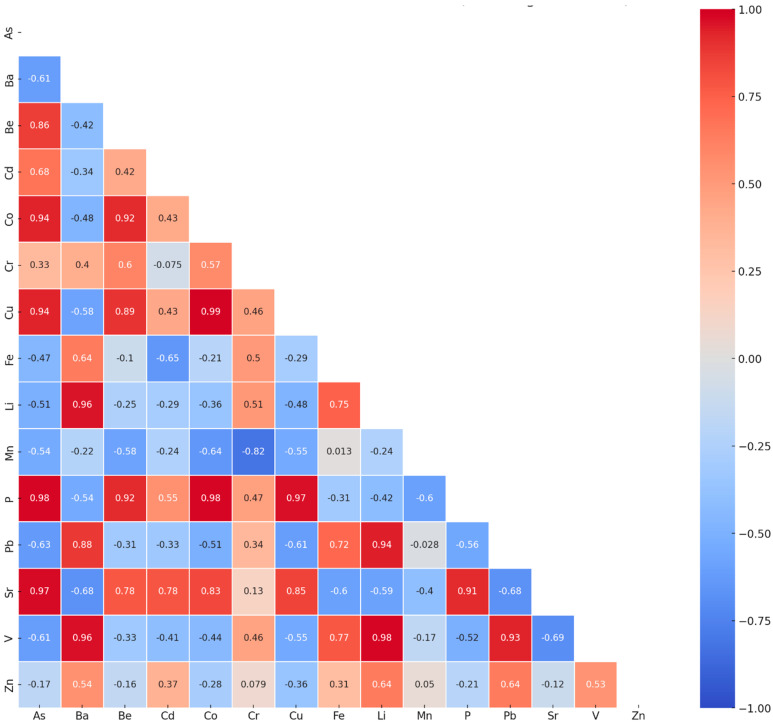
Lower half of the correlation matrix of element concentrations in samples from all collection points. The color scale used shows the maximum positive correlations (value of 1) in red and the maximum negative correlations (value of −1) in blue.

**Figure 4 biology-14-00222-f004:**
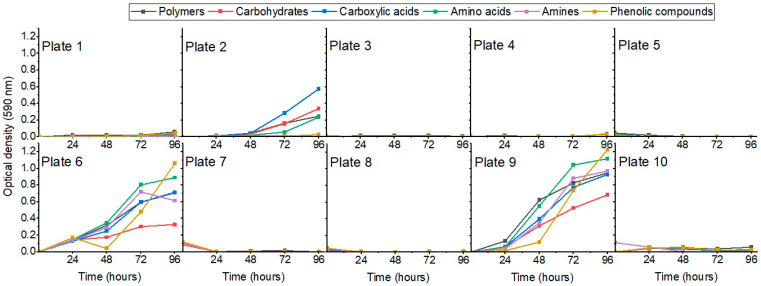
Average metabolic response for Biolog Ecoplates.

**Table 1 biology-14-00222-t001:** Sampling locations.

**Sample Number**	**Location Name**	**GPS Coordinates**
1	Camp Faraglione	74°42′57″ S–164° 06′ 41″ E Altitude 60 m asl
2	Camp Faraglione	
3	Camp Faraglione	
4	Camp Faraglione	
5	Edmonson Point	74°18′47.1″ S–165°04′10.2″ E Altitude 27 m asl
6	Edmonson Point	
7	Edmonson Point	
8	Apostrophe Island	73°31.134″ S–167°25.984″ E Altitude 37 m asl
9	Apostrophe Island	
10	Apostrophe Island	

**Table 2 biology-14-00222-t002:** Soil element content of sampling sites. Elements are expressed in mg/kg per dry mass.

Parameter	Unit	1	2	3	4	5	6	7	8	9	10
Dry matter	%	98.7	99.6	99.8	99.3	74.5	79	71.3	81.9	35.9	75
Ag	mg/kg	<0.50	<0.50	<0.50	<0.50	<0.50	<0.50	<0.50	<0.50	<0.66	<0.50
As	mg/kg	0.61	0.64	<0.50	<0.50	0.58	<0.50	<0.50	3.73	5.27	3.94
Ba	mg/kg	48.4	44.3	46.8	40.3	11.7	15.3	14	9.35	7.32	9.04
Be	mg/kg	0.193	0.199	0.15	0.227	0.151	0.167	0.185	0.28	0.269	0.273
Cd	mg/kg	<0.40	<0.40	<0.40	<0.40	<0.40	<0.40	<0.40	<0.40	<0.52	<0.40
Co	mg/kg	4.51	4.42	3.62	4.06	2	2.6	2.22	14.7	12.2	12.6
Cr	mg/kg	10.2	10.5	7.91	10.3	1.03	1.36	1.52	12.3	6.26	10.6
Cu	mg/kg	7.1	7.4	5.2	6.6	1.8	2.3	2	148	110	104
Fe	mg/kg	16,400	15,400	11,400	12,900	10,600	11,400	12,300	13,100	8,010	12,100
Hg	mg/kg	<0.20	<0.20	<0.20	<0.20	<0.20	<0.20	<0.20	<0.20	<0.26	<0.20
Li	mg/kg	29	27.7	17.9	21.4	2.7	3	3.3	3.2	1.9	2.7
Mn	mg/kg	176	180	142	152	218	228	256	154	150	133
Mo	mg/kg	<0.40	<0.40	<0.40	<0.40	0.54	0.52	0.66	<0.40	<0.52	<0.40
Ni	mg/kg	4.9	5.2	4.1	5.6	<1.0	1.2	1.3	65.9	65	60.3
P	mg/kg	793	795	504	634	426	521	451	2440	2640	2480
Pb	mg/kg	2.2	2.2	1.5	2.2	1.3	1.1	1.4	<1.0	<1.0	<1.0
Sb	mg/kg	<0.50	<0.50	<0.50	<0.50	<0.50	<0.50	<0.50	<0.50	<0.66	<0.50
Sn	mg/kg	<1.0	<1.0	<1.0	<1.0	<1.0	<1.0	<1.0	<1.0	<1.3	<1.0
Sr	mg/kg	10.2	10.6	9.14	10.6	13.5	19.8	15.1	39	70.7	45.9
Tl	mg/kg	<0.50	<0.50	<0.50	<0.50	<0.50	<0.50	<0.50	<0.50	<0.66	<0.50
V	mg/kg	20.5	19.6	15.3	18.1	6.63	9.62	7.42	7.83	4.28	6.17
Zn	mg/kg	35.7	33.4	23.9	25.8	22.8	22	26.3	17.4	32.4	19

## Data Availability

Data are available from the corresponding author upon reasonable request.

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
