# Peer review of "Antarctic Soil and Viable Microbiota After Long-Term Storage at Constant −20 °C"

_biology, 2025, doi:10.3390/biology14030222_

Round 1

Reviewer 1 Report

Comments and Suggestions for Authors

The manuscript of Pop et al. contains the characterisation of Antarctic soil samples after their storage for 13 years at -20oC. All the data on the composition of Antarctic microbial communities are very precious, as the access to these habitats demands significant efforts. Thus, the reviewed paper, definitely, deserves publication. However, many questions and comments should be answered/clarified before that.

1)    Authors present interesting data on the physicochemical characteristics of the soil samples. However, they are not discussed enough, neither in connection with microbial communities studied (arsenicum and the microorganism utilising it make the only exception), nor in the light of existing information obtained by other researchers. If such data have never been obtained previously, they should be compared with those of other environments, for example, Arctic permafrost, or any available data of the same kind. Is Antarctic permafrost soil different from other habitats and how it might influence the microbial communities – that’s the questions to answer. Of course, the discussion of the biological side of this question is most desirable, as the paper has been submitted to a biological journal.

2)    Significant attention, both in Results and in Discussion sections, is paid to positive and negative correlations between the abundance of different elements in the soil samples. However, these data, again, are not interpreted in terms of microorganisms present in these samples, while the Discussion section mostly repeats results. If there is no connection between these correlations and microorganisms found in the samples, these parts of text should be shortened or even excluded.

3)    For about 30 years microbial communities are characterised by culture-independent methods - the analyses of 16S rRNA genes in natural samples or, later, by metagenomic analyses. In this work authors studied the diversity of cultured bacteria, and this choice is partly justified, as it provides information on viable cells only. Still, the results depend a lot on the cultivation conditions, and here I have many concerns. (i) While the incubation temperature could be approved (15oC is the growth temperature of both obligate and facultative psychrophiles), the incubation time (4 days) is evidently too short (many years of storage at -20oC, slow growth of psychrophiles). (ii) Authors used commercial media, but it is worth presenting their composition. (iii) The utilization of chloramphenicol was really a surprise – why? Would it not be interesting to obtain as many growing bacteria as possible, not only chloramphenicol-resistant? All these points should be mentioned in the text and either justified, or, at least, discussed.

4)    The isolates obtained from agar plates were identified by 16S rRNA genes analyses, but no data on their level of similarity with the closest relatives are presented. Usually, 16S rRNA genes analyses allow us to identify microorganisms only on genus level, and comparison of genomes is needed to assign them to a certain species. However, if the level of similarity is 100%, the organism could be, with certain probability, identified on a species level. Authors should add a table to their text showing the % of 16S rRNA genes similarity of the strains in question with their closest relatives.

5)    There is also a question about a fungal isolate Geomyces pannorum – how it was identified, if 16S rRNA identification works only with prokaryotes?

More questions and comments:

Page 4, line 153   How microbial growth was determined?

Table 2     As and P are not metals, so either the title of the table should be corrected, or these two compounds should be presented in a separate table. By the way, are the quantitative data on these two compounds reliable if they were not supposed to be found?

Page 11, line 299      16S rRNA genes analyses

Page 11, lines 309-310    Not enough evidence on dormant stages – maybe these microorganisms continued multiplying with a very slow rate. Their very fast growth with almost no lag-phase does not support a dormant-stage version. I would advice to consider both possibilities, supporting them with references on the works of other authors.

Page 12, line 328      “where” instead of “were”

Page 12, line 336      the first report of its presence

Page 12, lines 337-339    Discussing the relation of isolates to temperature, authors should refer to characteristics of the most closely related microorganisms, as they did not perform any experiments with their isolates

Discussion. The order of subjects should be the same as in the results, so it should start with the physicochemical analyses of soil, and then move to microorganisms.

Conclusion is too general, and, if left at all, it should be made much more concrete.

Author Response

The manuscript of Pop et al. contains the characterisation of Antarctic soil samples after their storage for 13 years at -20oC. All the data on the composition of Antarctic microbial communities are very precious, as the access to these habitats demands significant efforts. Thus, the reviewed paper, definitely, deserves publication. However, many questions and comments should be answered/clarified before that.
 R: Dear Reviewer 1, thank you for your remarks and observation as it is an opportunity for us to improve the quality of our manuscript. Please see bellow a point-by-point reply, we hope that we managed to cover all the aspects mentioned. 

1)    Authors present interesting data on the physicochemical characteristics of the soil samples. However, they are not discussed enough, neither in connection with microbial communities studied (arsenicum and the microorganism utilising it make the only exception), nor in the light of existing information obtained by other researchers. If such data have never been obtained previously, they should be compared with those of other environments, for example, Arctic permafrost, or any available data of the same kind. Is Antarctic permafrost soil different from other habitats and how it might influence the microbial communities – that’s the questions to answer. Of course, the discussion of the biological side of this question is most desirable, as the paper has been submitted to a biological journal.
R: We acknowledge that the Discussion section was underdeveloped and we tried to complete it with the suggestions provided.

2)    Significant attention, both in Results and in Discussion sections, is paid to positive and negative correlations between the abundance of different elements in the soil samples. However, these data, again, are not interpreted in terms of microorganisms present in these samples, while the Discussion section mostly repeats results. If there is no connection between these correlations and microorganisms found in the samples, these parts of text should be shortened or even excluded.
R: Except for the Pseudomonas arsenicoxydans species, there is little possible correlation between the soil elements and microbiota, atleast from what we could find in the literature i.e. heavy metals contribution in formation of the microbial community (Lin et al. 2022) which has now been stated in discussion section. We have improved the discussion section but would like to keep the elements correlations analysis as it might be valuable for other researchers. Thank you for your understanding.

3)    For about 30 years microbial communities are characterised by culture-independent methods - the analyses of 16S rRNA genes in natural samples or, later, by metagenomic analyses. In this work authors studied the diversity of cultured bacteria, and this choice is partly justified, as it provides information on viable cells only. Still, the results depend a lot on the cultivation conditions, and here I have many concerns. (i) While the incubation temperature could be approved (15oC is the growth temperature of both obligate and facultative psychrophiles), the incubation time (4 days) is evidently too short (many years of storage at -20oC, slow growth of psychrophiles). (ii) Authors used commercial media, but it is worth presenting their composition. (iii) The utilization of chloramphenicol was really a surprise – why? Would it not be interesting to obtain as many growing bacteria as possible, not only chloramphenicol-resistant? All these points should be mentioned in the text and either justified, or, at least, discussed.
R: 
i) We used Biolog Ecoplates to evaluate the growth per substrate (well). From each well that we innoculated and that developed growth, we later put the content of the well on agar plates, isolated the colonies and sent the colonies to 16S analyses. As the Biolog Ecoplates manufacturer recommended, 4 days of incubation was performed, after about 2-3 days we had wells that where fully developed and other which presented no growth at all, and thus we set the deadline for 4 days, after 4 days some wells were fully developed (full substrate use) others had no growth or stagnant since day 3. 
Although we ceased to read the absorbance after 4 days, we continued to keep the Biolog Ecoplates plates in the incubator with the hope that other, more slowly growing strains will develop. Unfortunatelly, after about 7 days (11 in total) we’ve seen a contamination of all wells with Geomyces pannorum which started to spread out of control in all other wells within the plate, therefore we had to destroy the plates by autoclaving. We acknowledge the aspect raised but unfortunatelly only 4 days was possible with this method. Please see additional text added under “Microbial metabolic patern” section.
ii) Composition of the media used has now been mentioned in the manuscript at the materials and methods “Primary and final taxonomic identification” section.
iii) We used the chloramphenicol media only for fungus growth, and for its pasage, and thus to send a pure fungus sample to 16S service.

4)    The isolates obtained from agar plates were identified by 16S rRNA genes analyses, but no data on their level of similarity with the closest relatives are presented. Usually, 16S rRNA genes analyses allow us to identify microorganisms only on genus level, and comparison of genomes is needed to assign them to a certain species. However, if the level of similarity is 100%, the organism could be, with certain probability, identified on a species level. Authors should add a table to their text showing the % of 16S rRNA genes similarity of the strains in question with their closest relatives.
R: We attached the phylogenetic trees in supplementary materials, please see Supplementary 3. The value shown in the bracket represents the percent difference in the sequence alignment between the unknown and each individual library entry (please see explanation provided at the end of Supplementary 3 document), in example, the sample identified as Pseudomonas graminis has a 0.0 % difference in the sequence alignment compared to library entry, thus it was confirmed with 100%  certainly that it is indeed P. graminis; Geomyces pannorum has a 0.30% difference, Pseudomonas arsenicoxydans 0.10%, and Sporosarcina aquimarina a 1.59% difference. This data was now added in the manuscript as well.
 We also contacted via email the company from which we obtained the results, to get as many details as possible regarding their methods. 16S rRNA genes analyses data can now be found in materials and methods under “Primary and final taxonomic identification” section. We appologize for the inconvenience created but unfortunatelly due to technological limitations we do not have any way to perform these genetic analyses ourselves and had to contract it from a genetics service company (Accugenix Charles River, Germany), we hope that the information now provided is enough and suitable.

5)    There is also a question about a fungal isolate Geomyces pannorum – how it was identified, if 16S rRNA identification works only with prokaryotes?
 R: Data about the fungal isolate genetic analysis was also added. Please see “Primary and final taxonomic identification” section. “...the FunITS® analysis is the comparative sequencing of the ribosomal DNA ITS2 region in fungi…16S rRNA gene for bacteria (527R reverse and 005F forward primers) and the 350 bp ITS2 region for fungi (ITS3 forward and ITS4 reverse / ITS86F forward and ITS4 reverse primers) …”
More questions and comments:

Page 4, line 153   How microbial growth was determined?
R: The microbial growth in Biolog ecoplates is determined via absorbance at λ = 590 nm, higher absorbance represents higher microbial growth. We have now added this information clearly.

Table 2     As and P are not metals, so either the title of the table should be corrected, or these two compounds should be presented in a separate table. By the way, are the quantitative data on these two compounds reliable if they were not supposed to be found?
R: We appologize for that. The ICP OES analysis was performed by an acredited service (Measurlab Finland) and the elements are within their ISO method (raw data presented as supplementary in Supplementary table 1). We’ve changed the title of the table and performed the changes in text, thank you for this observation as well.

Page 11, line 299      16S rRNA genes analyses
R: corrected

Page 11, lines 309-310    Not enough evidence on dormant stages – maybe these microorganisms continued multiplying with a very slow rate. Their very fast growth with almost no lag-phase does not support a dormant-stage version. I would advice to consider both possibilities, supporting them with references on the works of other authors.

R: We refrain from making this statement as it should be the subject of a separate study, which we now take in consideration as we made glycerol stocks of the isolates for future research.

Page 12, line 328      “where” instead of “were”
R: corrected. Thank you.

Page 12, line 336      the first report of its presence
R: corrected

Page 12, lines 337-339    Discussing the relation of isolates to temperature, authors should refer to characteristics of the most closely related microorganisms, as they did not perform any experiments with their isolates
 R: We could not find relevant data on closely related strains in terms of cold resistance, however we provided information regarding cryo- and halophilic traits of G. pannorum [ref 40,41]. The detected S. aquimarina seem to be closely related to S. koreensis based on the neighbouring tree (please see Supplementary 3). In lines 452-457 we aimed at discussing distinctive relevant (cold-related) characteristics between the two.

Discussion. The order of subjects should be the same as in the results, so it should start with the physicochemical analyses of soil, and then move to microorganisms.
 R: the order has been changed as indicated.

Conclusion is too general, and, if left at all, it should be made much more concrete.
R: An improved conclusion section that states the highlights of the paper has been written.

We would like to thank you once again for your remarks and suggestions and we hope that we managed to succesfully implement them.

Kind regards,
The authors

Reviewer 2 Report

Comments and Suggestions for Authors

My comments are in attachment.

• What is the main question addressed by the research? First determination of chemical composition and microbiota of several antarctic soil samples collected in three different locations in Antarctica. • Do you consider the topic original or relevant to the field? Does it address a specific gap in the field? Please also explain why this is/ is not the case. The topic of this study is not original, but it is relevant to the field. Soil composition and microbiota are widely studied over the world, but not in Antarctica. Thus, the main novelty of this study is the information about chemical composition and microbiota of the soils collected at Antarctica continent. • What does it add to the subject area compared with other published material? Main novelty of this study is new location of the soil sample collection. Accordingly, the results obtained are unique. • Are the conclusions consistent with the evidence and arguments presented and do they address the main question posed? Please also explain why this is/is not the case. The conclusions are evidently consistent with the evidence and arguments presented. New information is obtained and new correlations between metal contents and microbiota are found. • Are the references appropriate? Yes. • Any additional comments on the tables and figures. No additional comments are on the tables and figures. The tables and figures are prepared with high accuracy.

Author Response

Dear Reviewer 2,

Thank you for taking the time to review our manuscript and for providing detailed suggestions for improving its clarity and readability. We appreciate your constructive feedback in helping us refine our work.

We have implemented the relevant changes to enhance the text quality and have provided a revised version of the manuscript. If there are any specific areas where further clarification is needed, we would be happy to address them.

Best regards,

The authors

Reviewer 3 Report

Comments and Suggestions for Authors

Reviewers comments on the Article Antarctic soil and microbiota after long-term storage at constant -20° C submitted to Biology MDPI

General comments:

Line 29: Please explain why and add to the text. Why samples were stored for 10 years, was it intentionally or not?

Line 43: Please inform readers shortly which m.o. grow on these 2 subzero temperatures without the need to go and read 2 different papers.

Some parts of the Introduction section remind of the Discussion section like lines 68 – 75. Please reorganize the Introduction section according to the publishers norms.

Line 141: Citation [21] leads to Phyton programming language page not to the scientific paper that explains how to calculate RMSD and dendrogram. This paper is not in the scope of computer science. Please share with the readers how RMSD and dendrogram were calculated and add the code to appendix if necessary.

Line 161: do you refer to the 16S small subunit ribosomal RNA? Please contact the external laboratory and provide in the text basic details of the molecular identification of bacteria: RNA isolation, primers, blasting etc. Same applies to lines 299-301

Line 293-296: Sentence belongs to Discussion section.

Line 305: “white-cream circular small and punctiform colonies”…

Please rephrase so it doesn’t sound like a cooking recipe.

Line 329: “μ550 g L−1 in sediments” - do you mean 550 μgL−1 in sediments?

Sentence “Unlike Sporosarcina antarctica which was discovered in 2008 from soil samples collected off King George Island, West Antarctica [38], this is yet again the first report of presence in Antarctic soil” sounds very confusing, please reformulate for the readers convenience.

Sentence “The strains of microorganisms identified in this work seem to be rather facultative psychrophilic, than real psychotrophs, and they are oligotrophic, adapted to both mild and harsh environmental conditions. Their resistance to -20° C for so many years can be appealing for future extreme conditions environmental studies” suggests that normal temperatures in the Antarctic are above -20 degrees. Is this so? Can you please provide average year temperature in the isolation site.  

Please provide in the Discussion section more detailed information and description of all strains that you identified in this work.

Discussion section needs more discussion and comparing to already published papers, that will also extend the list of references.

Conclusion is short and inconclusive. Please use this section to provide the conclusion and recapitulation of your findings and their importance.

Author Response

Dear Reviewer 3,

Thank you for your constructive remarks, please see bellow our point-by point replies.

Reviewers comments on the Article Antarctic soil and microbiota after long-term storage at constant -20° C submitted to Biology MDPI

General comments:

Line 29: Please explain why and add to the text. Why samples were stored for 10 years, was it intentionally or not?

R: The samples were stored for analysis but unfortunatelly due to the death of researcher responsible –Teodor Negoita (mentioned in Acknowledgemens section, to whom the work is dedicated), they remained stored in cold until granted to us by the keeper in 2024. We’ve added this information at line 29 as requested, and under materials and methods.

Line 43: Please inform readers shortly which m.o. grow on these 2 subzero temperatures without the need to go and read 2 different papers.

R: Thank you for this observation as well, we’ve added the information please see lines 48-50..

Some parts of the Introduction section remind of the Discussion section like lines 68 – 75. Please reorganize the Introduction section according to the publishers norms.

R: We moved the sentence to Discussion section and further developed the Introduction section.

Line 141: Citation [21] leads to Phyton programming language page not to the scientific paper that explains how to calculate RMSD and dendrogram. This paper is not in the scope of computer science. Please share with the readers how RMSD and dendrogram were calculated and add the code to appendix if necessary.

R: Even if the RMSD comparison reveals the global similarity between two spectra, the analysis is not usually applied to spectra. Thus, we applied a slightly different analysis based on Euclidean distances, which appears more sensitive to overall spectral shape and for which we found citations. As a consequence, we revised the method section and Figure 1.

Line 161: do you refer to the 16S small subunit ribosomal RNA? Please contact the external laboratory and provide in the text basic details of the molecular identification of bacteria: RNA isolation, primers, blasting etc. Same applies to lines 299-301

R: We’ve contacted the service supplier via email multiple times and gathered as much data as they could provide us (some details were confidential), the information is now in the manuscript under “Primary and final taxonomic identification” section, lines 193-208. Thank you for your understanding.

Line 293-296: Sentence belongs to Discussion section.

R: the sentence has been moved to the Discussion section

Line 305: “white-cream circular small and punctiform colonies”…

Please rephrase so it doesn’t sound like a cooking recipe.

R: We’ve reformulated the descriptions in a clearer manner.

Line 329: “μ550 g L−1 in sediments” - do you mean 550 μgL−1 in sediments?

R: We apologize for this as well, we have corrected the error.

Sentence “Unlike Sporosarcina antarctica which was discovered in 2008 from soil samples collected off King George Island, West Antarctica [38], this is yet again the first report of presence in Antarctic soil” sounds very confusing, please reformulate for the readers convenience.

R: The paragraph was reformulated to be clearer.

Sentence “The strains of microorganisms identified in this work seem to be rather facultative psychrophilic, than real psychotrophs, and they are oligotrophic, adapted to both mild and harsh environmental conditions. Their resistance to -20° C for so many years can be appealing for future extreme conditions environmental studies” suggests that normal temperatures in the Antarctic are above -20 degrees. Is this so? Can you please provide average year temperature in the isolation site.  

R: We now cited and provided official data regarding temperatures, we did not intend to suggests that normal temperatures in the Antarctic are above -20 degrees and we appologize for any confusion that might have been created.

Please provide in the Discussion section more detailed information and description of all strains that you identified in this work.

R: Additional details regarding the identified strains has been provided

Discussion section needs more discussion and comparing to already published papers, that will also extend the list of references.

R: We have improved the discussion section, please see the revised section within the manuscript.

Conclusion is short and inconclusive. Please use this section to provide the conclusion and recapitulation of your findings and their importance.

R: A more representative conclusion section that states the highlights of the paper has been written.

The authors would like to thank Reviewer 3 once again for the remarks and suggestions provided, and we hope that we managed to improve the quality of the manuscript.

Best regards,

The authors

Reviewer 4 Report

Comments and Suggestions for Authors

Comments to manuscript: Antarctic soil and microbiota after long-term storage at constant 2 -20° C, submitted by Pop et al.

The authors investigated soil samples collected at different sites of East continental Antarctica after more than 10 years of storage. They analyzed the soils for metals and organic matter, using inductively coupled plasma atomic emission spectroscopy and Fourier transform infrared spectroscopy. They isolated four microorganisms from the soils by use of Biolog Ecoplates (no reference), which is known to be difficult (Sofo and Ricciuti, 2019). Based on these experiments, four organisms were identified. Pop et al. determined polymers, carbohydrates, carboxylic acids, amino acids, amines and phenolic compounds to describe the “average metabolic response” (response to what?). The four organisms were externally identified by a company.

The focus of the manuscript is clearly the metal and organic matter analysis of the soil samples, which is, however, uncomplete. As can be seen in Tab. 2, important constituents were not determined, such as N or Mg, no pH is given.

Regarding the microorganisms, the isolation is incompletely described, the organisms were not studied for properties which enabled them to survive under the extreme conditions. What means “Average metabolic response”? How were 2-hydroxy benzoic acid and 4-hydroxy benzoic acid identified?  Which microorganism was cultured on which plate (Fig. 2)? The sequences for species determination are not even presented as supplementary data. Altogether, only three bacteria and the fungus Geomyces pannorum were identified. These species are existing also in completely different habitats in other parts of the world. There might be strain-specific differences in survival strategies and metabolic properties, therefore, the entire genomes should be sequenced to characterize these differences, compared to strains from other regions.  This was not addressed.  There is also no DNA extraction of the soil for NGS, which could give hints for a larger richness of the original population of microorganisms, especially as there seem to be no measurements, formerly done with the fresh samples (control). A correlation with the different soil samples is therefore not possible. In summary, the biological part of the manuscript is by far too poor, and the soils are not sufficiently characterized.

Author Response

Dear Reviewer 4,

Thank you for your constructive remarks and suggestions.

Please find bellow a point-by-point reply.

Comments to manuscript: Antarctic soil and microbiota after long-term storage at constant 2 -20° C, submitted by Pop et al.

The authors investigated soil samples collected at different sites of East continental Antarctica after more than 10 years of storage. They analyzed the soils for metals and organic matter, using inductively coupled plasma atomic emission spectroscopy and Fourier transform infrared spectroscopy. They isolated four microorganisms from the soils by use of Biolog Ecoplates (no reference), which is known to be difficult (Sofo and Ricciuti, 2019). Based on these experiments, four organisms were identified. Pop et al. determined polymers, carbohydrates, carboxylic acids, amino acids, amines and phenolic compounds to describe the “average metabolic response” (response to what?). The four organisms were externally identified by a company.

R: Please allow us to split the comment in two parts for a better resolution of our answer.

  1. i) “The authors investigated soil samples collected at different sites of East continental Antarctica after more than 10 years of storage. They analyzed the soils for metals and organic matter, using inductively coupled plasma atomic emission spectroscopy and Fourier transform infrared spectroscopy. They isolated four microorganisms from the soils by use of Biolog Ecoplates (no reference), which is known to be difficult (Sofo and Ricciuti, 2019).”

Citations of the method Biolog Ecoplates have now been added. We aimed at evaluating live cells only, thus this method seemed as one of the most suitable. In example, Tobias et al., 2021 [now ref #23] evaluated the community level physiological profiles (CLPP) of bacterial communities inhabiting polluted environments using Biolog Ecoplates, in comparison to non-contaminated sites. Other authors (Lukhele et al., 2021) [now ref# 24] mentioned that previous studies did not consider both prokaryotic and eukaryotic communities when linking metabolic potential and activity, community composition, and environmental gradients. To address this gap, they profiled microbial functional potential using the Biolog Ecoplates method.

We have added these two citations and provided clarifications.

  1. ii) “Based on these experiments, four organisms were identified. Pop et al. determined polymers, carbohydrates, carboxylic acids, amino acids, amines and phenolic compounds to describe the “average metabolic response” (response to what?). The four organisms were externally identified by a company.”

We did not determine polymers, carbohydrates, carboxylic acids, amino acids, amines and phenolic compounds, these compounds are the substrates of the wells of the Biolog Ecoplate, as stated in Microbial metabolic pattern section : “The Biolog Ecoplates are 96-wells plates that hold 31 carbon assays in triplicate, which are split in 6 groups: polymers, carbohydrates, carboxylic acids, amino acids, amines and phenolic compounds.“. We appologize for any confusion created.

The microbial growth that developed was quantified as “average metabolic response” this is the manufacturer proposed method, and also reported by other authors, please see Xiong et al., 2012 : “The average metabolic response (AMR) of the soil samples was calculated as the average of the mean difference between the O.D. of the carbon source containing wells and the control wells“, also Preston-Mafham et al., 2002 provides a good review report regarding the method. Both citations have now been added [ ref # 32 and 33].

The organisms that developed were plated and isolated, and then the colonies were sent to an acredited service provider as due to technological limitation we can not perform such analysis.

The focus of the manuscript is clearly the metal and organic matter analysis of the soil samples, which is, however, uncomplete. As can be seen in Tab. 2, important constituents were not determined, such as N or Mg, no pH is given.

R: Thank you for this observation, we aimed at evaluating the live microbiota as well as the soil elements, the ICP-OES analysis unfortunatelly could not determine N and Mg and we appologize for not thinking about evaluating the pH parameter which could have provided valuable data in terms of soil quality and as a possible microbiota community influence factor.

Regarding the microorganisms, the isolation is incompletely described, the organisms were not studied for properties which enabled them to survive under the extreme conditions. What means “Average metabolic response”? How were 2-hydroxy benzoic acid and 4-hydroxy benzoic acid identified?  Which microorganism was cultured on which plate (Fig. 2)? The sequences for species determination are not even presented as supplementary data. Altogether, only three bacteria and the fungus Geomyces pannorum were identified. These species are existing also in completely different habitats in other parts of the world. There might be strain-specific differences in survival strategies and metabolic properties, therefore, the entire genomes should be sequenced to characterize these differences, compared to strains from other regions.  This was not addressed.  There is also no DNA extraction of the soil for NGS, which could give hints for a larger richness of the original population of microorganisms, especially as there seem to be no measurements, formerly done with the fresh samples (control). A correlation with the different soil samples is therefore not possible. In summary, the biological part of the manuscript is by far too poor, and the soils are not sufficiently characterized.

R: We have added the necessary details regarding the strain isolation and genetic analysis performed to identify the species, please see “Primary and final taxonomic identification” section.

No soil DNA extraction was performed for NGS as we did not perform this analysis nor had we intended, as the purpose of the work was to evaluate viable microorganisms only.

Thank you once again for your thorough and insightful review. We appreciate the time and effort you have taken to evaluate our work and your comments have been valuable in helping us improve the overall quality of the manuscript.

Best regards,

The authors

Round 2

Reviewer 3 Report

Comments and Suggestions for Authors

No more comments.

Author Response

Thank you for your remarks and suggestions

Reviewer 4 Report

Comments and Suggestions for Authors

The authors improved their manuscript.  I have only one suggestion: Pseudomonas arsenicoxydans is only one possibility of several others, according to NCBI blasting. Others with the same max score and query cover are Pseudomonas sp., P. proseti, P. lini, and uncultured bacterium.  Please describe why you are concerned that  bacterium  P. arsenicoxydans is the correct species, or write Pseudomonas sp. instead.
